# Development of Embryo Suspensors for Five Genera of Crassulaceae with Special Emphasis on Plasmodesmata Distribution and Ultrastructure

**DOI:** 10.3390/plants9030320

**Published:** 2020-03-03

**Authors:** Małgorzata Kozieradzka-Kiszkurno, Daria Majcher, Emilia Brzezicka, Joanna Rojek, Justyna Wróbel-Marek, Ewa Kurczyńska

**Affiliations:** 1Department of Plant Cytology and Embryology, Faculty of Biology, University of Gdańsk, Wita Stwosza 59, 80-308 Gdańsk, Poland; daria.czaplejewicz@gmail.com (D.M.); emka2109@wp.pl (E.B.); joanna.rojek@ug.edu.pl (J.R.); 2Institute of Biology, Biotechnology and Environmental Protection, Faculty of Natural Sciences, University of Silesia, Jagiellońska 28, 40-032 Katowice, Poland; justyna.wrobel@us.edu.pl (J.W.-M.); ewa.kurczynska@us.edu.pl (E.K.)

**Keywords:** *Aeonium*, *Aichryson*, *Echeveria*, embryo, embryogenesis, endopolyploid cells, *Monanthes*, ovule, *Sedum*, symplasmic connection

## Abstract

The suspensor in the majority of angiosperms is an evolutionally conserved embryonic structure functioning as a conduit that connects ovule tissues with the embryo proper for nutrients and growth factors flux. This is the first study serving the purpose of investigating the correlation between suspensor types and plasmodesmata (PD), by the ultrastructure of this organ in respect of its full development. The special attention is paid to PD in representatives of Crassulaceae genera: *Sedum*, *Aeonium*, *Monanthes*, *Aichryson* and *Echeveria*. The contribution of the suspensor in transporting nutrients to the embryo was confirmed by the basal cell structure of the suspensor which produced, on the micropylar side of all genera investigated, a branched haustorium protruding into the surrounding ovular tissue and with wall ingrowths typically associated with cell transfer. The cytoplasm of the basal cell was rich in endoplasmic reticulum, mitochondria, dictyosomes, specialized plastids, microtubules, microbodies and lipid droplets. The basal cell sustained a symplasmic connection with endosperm and neighboring suspensor cells. Our results indicated the dependence of PD ultrastructure on the type of suspensor development: (i) simple PD are assigned to an uniseriate filamentous suspensor and (ii) PD with an electron-dense material are formed in a multiseriate suspensor. The occurrence of only one or both types of PD seems to be specific for the species but not for the genus. Indeed, in the two tested species of *Sedum* (with the distinct uniseriate/multiseriate suspensors), a diversity in the structure of PD depends on the developmental pattern of the suspensor. In all other genera (with the multiseriate type of development of the suspensor), the one type of electron-dense PD was observed.

## 1. Introduction

The formation of viable seeds in flowering plants depends on the correct ovule development before and after fertilization. Upon fertilization, the synchronous growth and development of the embryo, the endosperm and the seed coat require communication between these tissues [1,2]. Embryogenesis in most angiosperms is initiated by the process of double fertilization, in which one sperm cell fertilizes the egg cell to form the zygote, and the other fertilizes the central cell to form the endosperm [3]. The zygote undergoes asymmetric cell division, giving rise to two daughter cells with distinct developmental fates. The embryo proper is generated through a cell lineage founded by a small apical (chalazal) cell, while the embryo-suspensor, a terminally differentiated structure, is formed through a cell lineage established by a larger, basal (micropylar) cell. Many studies on embryo development in several plant species have shown that the suspensor is a unique embryonic region that connects the embryo to the maternal (i.e., ovule) tissues, and that compared to the embryo proper, it is morphologically distinct throughout the plant kingdom [4,5,6]. The suspensor exhibits considerable variation with regard to its shape, number of cells, size and ploidy level [7]. In many flowering plants, suspensor cell differentiation is accompanied by endoreduplication leading to a high degree of ploidy [8]. It seems that endopolyploid cells are particularly important for tissues and organs that actively function for only a short period of development [9]. The suspensor is a short-lived organ which is eliminated by programmed cell death (PCD) during late embryogenesis and does not contribute to the next plant generation [10]. The short life cycle of this organ in most plants further stresses its role in the nourishment, growth and differentiation of young embryos. Structural and experimental findings clearly reveal that this organ acts as a conduit for nutrient flow and may deliver unique metabolites to enable the growth of the embryo proper [4,11]. In addition, the contribution of the suspensor to transport nutrients to the embryo is also confirmed by the ultrastructure analysis of the organ, especially the development of walls that are typically used for the so-called transfer cells that are specialized in short-distance, active movement of substances through plasmalemma [12,13]. Many suspensors contain giant basal cells at their micropylar end, which is the site of maximum metabolic activity [6]. In some angiosperm members, for example, Crassulaceae, haustorial protrusions that penetrate the ovular tissues are produced by the suspensor basal cell, and these protrusions are the main source of nutrients for the embryo proper [14,15,16]. The family Crassulaceae has the most number of species among the Saxifragales and most of its members are leaf succulents [17]. This taxonomically complex angiosperm family comprises approximately 1400 species currently classified in 35 genera [18]. Their highly diverse morphology, cytology and habit complicate the systematics of the family Crassulaceae and the relationships between species and genera remain unclear [19]. Additionally, difficulty in the reconstruction of relationships between members of the family Crassulaceae is related with the homoplasy of phenotypic characters noted within family representatives [19,20]. Even though the genera show global presence, they mostly occur in the Northern Hemisphere and southern Africa, typically in dry and/or cold areas with scarce water [17].

Crassulaceae are also an interesting topic of study in terms of embryology, mainly with regard to their large polyploid suspensors with haustoria [21]. Nearly eighty-five years ago, Mauritzon (1933) described the structural aspects of embryogenesis in several genera of Crassulaceae (among others, *Aeonium*, *Crassula*, *Echeveria*, *Rosularia*, *Kalanchoe* and *Monanthes*) but only at the level of the light microscope. However, previous data on embryogenesis are primarily schematic illustrations that were drawn from microscopic observations and show some stages of embryogenesis in selected species. Until now, most of the cytochemical and ultrastructural studies of the suspensor in this family have been performed on the *Sedum* genus (the most species-rich member of the family) and mainly with *S*. *acre* and *S*. *hispanicum* [22,23], *S*. *reflexum* [16] and to a lesser degree in *Sempervivum arachnoideum*, *Jovibarba sobolifera* and *Graptopetalum bellum* [15]. In *Sedum* special attention has been paid to studies on the ultrastructure of the compound plasmodesmata during the development of the suspensor [24]. Atypical compound plasmodesmata containing electron-dense material in the suspensor have been found in species from the genus *Sedum* as well as in species from other Crassulaceae genera: *Sempervivum arachnoideum* and *Jovibarba sobolifera* [25]. These complex cytoplasmic bridges are wider than normal ones and electron-dense material associated with these plasmodesmata has continuity with profiles of the rough endoplasmic reticulum. Since their discovery, plasmodesmata have been the focus of intense investigation in our laboratory. Our further studies on Crassulaceae revealed that the walls between the suspensor basal cell and the endosperm also contain the same type of the compound plasmodesmata as that found in the suspensor [26]. The wall suspensors of a few species of Fabaceae, namely *Glycine* [27], *Medicago* [28] and *Vicia faba* [29] exhibited a similar distribution of plasmodesmata. Recent studies on *Sedum acre* in Crassulaceae have analyzed symplasmic communication between the basal cell and the embryo proper and endosperm. These studies showed that symplasmic communication is nonuniform [30]. Indeed, despite many important studies about the embryogenesis of Crassulaceae, there is a lack of a detailed description of ultrastructural aspects for a large number of genera from this family. Therefore, the selection of our plant material is not accidental. In this report, we are extending the scope of our research to include other genera not previously described. In addition, two species of *Sedum* were analyzed, which had not been tested before.

This is the first ultrastructural analysis of the suspensor in selected species from five genera of Crassulaceae: *Sedum*, *Aeonium*, *Monanthes*, *Aichryson* and *Echeveria*. The main goal of the present study was an examination of the correlations between suspensor types and plasmodesmata structure (in the case of the two species of *Sedum* with distinct uniseriate or multiseriate suspensors). We also determined whether all genera/species with a morphologically similar (multiseriate) suspensor have plasmodesmata that are not only the same structurally, but also distributed in the same way.

Expanding knowledge about Crassulaceae embryology and ultrastructure plasmodesmata allows comparison of collected data between other (not studied) genera/species which may provide new and taxonomically useful knowledge. Our findings also enable better understanding of the evolutionary process of the ovule-embryo relationship establishing in Crassulaceae.

## 2. Results

Our embryological studies show the occurrence of diverse suspensor morphologies in representatives of five genera of Crassulaceae. In all species, the morphology and ultrastructure of the suspensor was investigated during full development and functioning.

In *Sedum*, two different types of suspensor development were observed: a long uniseriate (Figure 1A) and a few-celled multiseriate (Figure 1C).

Generally, a uniseriate suspensor was made up of the basalmost cell (micropylar) and a single file of 7 to 10 additional cells (referred to as the chalazal suspensor cells). In all species studied here, the morphology and ultrastructure of the basalmost cell is clearly distinct from all the others suspensor cells and the term ‘basal cell’ has been used to determine this cell. The basal cell develops the micropylar haustorium which penetrates the maternal tissues (Figure 1A,B). This type of suspensor is described only in *Sedum sediforme*.

However, a few-celled multiseriate suspensor occurs in *S*. *atratum* and other genera studied and consists of an enlarged basal cell forming a suspensor haustorium and two or four chalazal suspensor cells in two layers (Figure 1C,D).

### 2.1. Uniseriate Suspensor Morphology

#### *Sedum sediforme* (Figure 2A)

This species has an elongate and filamentous suspensor. The fully developed suspensor consists of a uniseriate file of 7–10 cells. The slightly elongated basal cell (~40 × 25 μm) forms a 1-celled suspensor haustorium which is already well developed and ramifies in the integumentary tissue. The chalazal suspensor that connects the basal cell to the embryo proper consists of highly elongated and vacuolated cells. The cellular endosperm surrounds the suspensor and embryo proper. Only in this species are the micropylar endosperm cells large in size, have an irregular shape and squeeze in between the cells of surrounding tissues. The suspensor is three to four times longer than the embryo proper.

### 2.2. Few-Celled Multiseriate Suspensor Morphology

#### 2.2.1. *Sedum atratum* (Figure 2B)

The suspensor morphology in this species varies substantially from that found in *S*. *sediforme*. Namely, the suspensor is a few-celled multiseriate. This embryological organ is built of a large pear-shaped basal cell (~100 × 60 μm) and two to four chalazal suspensor cells of rectangular shape in two layers. A branched haustorium is well developed and penetrates the integuments and the funiculus. The endosperm completely surrounds the suspensor cells.

#### 2.2.2. *Aeonium sedifolium* (Figure 2C)

The fully developed suspensor consists of a very large spherical basal cell (~80 × 70 μm) and a few chalazal suspensor cells in two layers. The haustorium is quite underdeveloped. The endosperm cells closely surround the suspensor cells.

#### 2.2.3. *Monanthes anagensis* (Figure 2D)

The suspensor contains a greatly elongated basal cell (~70 × 20–30 μm) with a strongly developed micropylar haustorium and a few chalazal suspensor cells of rectangular shape in one layer. The cellular endosperm cells surrounds both the suspensor and the embryo proper.

#### 2.2.4. *Aichryson laxum* (Figure 2E)

The suspensor consists of a huge basal cell (~25 × 25–30 μm) and a few chalazal suspensor cells in two layers. The micropylar haustorium is poorly developed. The endosperm cells closely surround the suspensor.

#### 2.2.5. *Echeveria lutea* (Figure 2F)

The completely developed suspensor is built from a large pear-shaped basal cell (~100 × 40–50 μm) and 2–4 chalazal suspensor cells in two/three layers. The haustorium is moderately developed and penetrates the integumentary tissues. The cellular endosperm surrounds the suspensor cells.

### 2.3. Ultrastructure of the Suspensor

#### 2.3.1. *Sedum sediforme* (Figure 3)

The basal cell contains a huge single nucleus which is the most prominent organelle and is located in the micropylar part of the cell. The enlarged nucleus moves from the basal cell towards the micropylar haustorium during development (Figure 3A). The basal cell cytoplasm has a variety of different organelles: mitochondria, plastids, dictyosomes, profiles of endoplasmic reticulum (ER) and vacuoles. Mitochondria of different size (both spherical and ellipsoidal) are distributed throughout the cytoplasm and both have a few tubular cristae (Figure 3B,C). Plastids are generally larger than mitochondria and they are primarily concentrated near the nucleus. Furthermore, most of these plastids are spherical and electron-dense and contain many vesiculate lamellae and small tubules. Different sized vacuoles contain fine fibrillar material (Figure 3B,C). The endosperm is embedded completely in the suspensor cells (Figure 3B,C). The micropylar endosperm walls form an extensive network of ingrowths (Figure 3B).

The density of the basal cell cytoplasm differs from that of the chalazal suspensor cells cytoplasm (Figure 3D). In the cytoplasm of the first cell of the chalazal suspensor cells (the most proximal to the basal cell), some rough endoplasmic reticulum (RER) cisternae appear to be stacked in three to four layers. Furthermore, many mitochondria are distributed throughout the cytoplasm (Figure 3C,D). Fine transfer-wall ingrowths are formed by the wall separating the basal cell from the first layer of the chalazal suspensor cells. Numerous mitochondria, spherical to ellipsoidal in shape, are present between the transfer wall ingrowths (Figure 3C,D). All of the lateral (on the border with the endosperm) (Figure 3C) and transverse walls of the suspensor cells have simple plasmodesmata (Figure 3D,E). The chalazal cells of the suspensor are strongly vacuolated (Figure 3E), with the exception of the first cell bordering the basal cell (Figure 3C,D). Some lipid droplets are also seen in the cytoplasm of all chalazal suspensor cells (Figure 3D). There are a few electron-dense plastids and some of these may contain starch grains (Figure 3C,E). The dense cytoplasm of the haustorium contains: mitochondria, ER, a few electron-dense plastids and vacuoles (Figure 3F).

#### 2.3.2. *Sedum atratum* (Figure 4)

The suspensor structure in this species varies substantially from that found in *S*. *sediforme*. Compared to *S*. *sediforme*, the number and size of wall ingrowths increase at the micropylar apex of the basal cell (Figure 4A).

Ultrastructural examination reveals fine ingrowths formed by the micropylar part of the suspensor basal cell wall. Mitochondria occur singly in the cytoplasm and are located near the wall ingrowths (Figure 4A). The basal cell cytoplasm in this region is filled with short profiles of ER, dictyosomes producing many vesicles. In addition, a few small myelin-like bodies are present in the cytoplasm (Figure 4A). A single large nucleus containing one dense nucleolus is located in the central part of the basal cell (Figure 4B). Numerous branched plasmodesmata with electron-dense material on the basal cell side perforate the wall between the suspensor basal cell and the endosperm cells (Figure 4B). Numerous microtubules are readily detected near the surface of the nuclear envelope (Figure 4B, inset). In the vicinity of the nucleus, the cytoplasm contains very numerous long profiles of endoplasmic reticulum. Plastids are not very numerous. Ultrastructurally, these are similar to those observed in *S. sediforme*, although their shape is more irregular (Figure 4C). Branched plasmodesmata with electron-dense material on the basal cell side perforate the wall separating the basal cell from the first layer of the chalazal suspensor cells (the most proximal to the basal cell) (Figure 4D,E).

The cytoplasm of the chalazal suspensor cells is filled with: mitochondria, profiles of ER, plastids, microbodies and vacuoles containing fine fibrillar material. There are some lipid droplets in the cytoplasm of these cells (Figure 4D,E).

A branched micropylar haustorium invades integumentary tissues. In the haustorium almost all the space between extensive wall ingrowths is filled with a very large number of mitochondria (Figure 4F).

#### 2.3.3. *Aeonium sedifolium* (Figure 5)

Fine ingrowths are formed by the micropylar apex of the basal cell wall (Figure 5A,B). Generally, the wall ingrowths have a typical ultrastructure for transfer cells; the plasmalemma closely follows the contours of the ingrowths (Figure 5B). Many spherical- to ellipsoidal-shaped mitochondria are present between the transfer wall ingrowths (Figure 5C,D). The basal cell cytoplasm contains various organelles: mitochondria, plastids, dictyosomes, profiles of ER, microbodies, vacuoles and vesicles differing in size and content (Figure 5B,C). Mitochondria are the most numerous of the organelles and occur throughout the cytoplasm. They have well-developed cristae. The dense basal cell cytoplasm contains many ribosomes that may be free or attached to membranes of the RER. The basal cell cytoplasm has a well-developed smooth endoplasmic reticulum (SER). Uniform distribution of single cisternae of the RER is observed throughout the basal cell (Figure 5C). Dictyosomes are evenly distributed within the cytoplasm of the suspensor basal cell. Numerous vesicles appear to have originated from dictyosomes. In addition, a few small microbodies are present in the cytoplasm (Figure 5B,C). A few plastids are observed in the basal cell cytoplasm (Figure 5C). A single large nucleus containing one dense nucleolus is located in the central part of the basal cell. Microtubules are concentrated near the nucleus (Figure 5C). Similar to that observed in *S*. *atratum*, the transverse wall separating the basal cell from the first layer of chalazal cells (the most proximal to the basal cell) is arched and perforated by plasmodesmata with electron-dense material on the cell side, but the plasmodesmata are not branched (Figure 5D,E). Similar plasmodesmata are also observed in the wall between the basal cell and the endosperm cells (not shown).

The chalazal suspensor cells have much smaller plastids than the basal cell. Some of the plastids of the chalazal suspensor cells are cup-shaped, with invaginations of cytoplasm that are less dense than the cytoplasm surrounding the plastid (Figure 5D). There are a large number of plasmodesmata in the inner walls of the chalazal suspensor (Figure 5D,E).

In the haustorium, fine wall ingrowths are formed. Mitochondria are distributed between wall ingrowths. Plastids and dictyosomes are also visible in the haustorium cytoplasm (Figure 5F).

#### 2.3.4. *Monanthes anagensis* (Figure 6)

The basal cell contains a huge single nucleus which is the most prominent organelle and is located in the chalazal part of the cell (Figure 6A). At the tip of the basal cell prior to intrusion through the integument, the cell has a very dense cytoplasm with few organelles such as dictyosomes, mitochondria and small vacuoles. A few profiles of the ER are oriented parallel to the long axis of the basal cell (Figure 6B). The micropylar haustorium of the basal cell is already well developed and ramifies in the integumentary tissue. Fine ingrowths are formed by the micropylar haustorium wall. Generally, the cytoplasm of the haustorium basal cell is rich in mitochondria, profiles of ER and dictyosomes (Figure 6B). The basal cell is filled with dense cytoplasm containing numerous ribosomes lying free or covering outer membranes of rough ER, dictyosomes, mitochondria, microbodies, plastids, vacuoles and vesicles differing in size and content. The profiles of the endoplasmic reticulum are rather long and may be scattered singly throughout the cytoplasm (Figure 6C,D). The ER frequently runs parallel to the plasmalemma (Figure 6B) and to the surface of the plastids (Figure 6C). Some rough ER cisternae appear to be stacked in three to four layers parallel to long axis of the cell or the nuclear surface (Figure 6C). Microbodies occur less frequently in the cytoplasm than other organelles (not shown). The mitochondria are spherical to ellipsoidal in shape. They have the highest abundance among all the organelles and occur throughout the cytoplasm. They have well-developed cristae (Figure 6C,D). Plastids are considerably larger than the mitochondria, and sometimes are variable in size and shape with an irregular outline. They are not very numerous in the cytoplasm. These organelles are mainly situated near the nucleus. The stroma of the plastids is electron-dense and consists of small tubules or lamellae. The peripheral reticulum is well developed. Occasionally, electron-translucent inclusions are also present in the plastids (Figure 6C,D).

The wall between the basal cell and the first layer of the chalazal suspensor cells (the most proximal to the basal cell), contains unbranched plasmodesmata with electron-dense material, which appears from the basal cell side (Figure 6E,F). Similar plasmodesmata are also observed in the wall between the basal cell and the endosperm cells (Figure 6E). There are simple plasmodesmata in the inner walls of the chalazal suspensor and embryo proper (Figure 6G). The chalazal suspensor consists of 2–3 cells in one layer. Mitochondria are not very numerous; they are spherical or ellipsoidal in shape. The few plastids are generally smaller than the basal cell plastids. The size and structure of the spherical nucleus of the chalazal suspensor cells do not differ from those of the nuclei of the embryo proper. Numerous vacuoles are bigger than the basal cell vacuoles (Figure 6E).

#### 2.3.5. *Aichryson laxum* (Figure 7)

Fine ingrowths are formed by the micropylar part of the basal cell wall (Figure 7A). The huge basal cell includes a single large and irregularly shaped nucleus with one nucleolus, which is located in the central part of the basal cell. The nucleus is the most prominent organelle within the cell (Figure 7A,C). Many mitochondria of different size (both spherical and ellipsoidal) are distributed throughout the cytoplasm of the basal cell and both have a few tubular cristae (Figure 7A,B). Dictyosomes are evenly distributed within the cytoplasm of the suspensor cell. The ER frequently runs parallel to the plasmalemma. Some rough ER cisternae appear to be stacked in three to four layers parallel to the cell or the nuclear surface. Microbodies occur less frequently in the cytoplasm than other organelles (Figure 7A,B). The plastids in the basal suspensor cell are not very numerous. They are conspicuous in the chalazal part of the basal cell, and contain an electron-dense stroma and a few internal lamellae. The peripheral reticulum is clearly visible. These organelles are mainly situated near the nucleus. Notably, there are single lipid droplets (Figure 7C).

In the transverse cell which separates the basal cell from the chalazal cells, there are many plasmodesmata with electron-dense material (Figure 7C,D). Occasionally, similar plasmodesmata also observed in the wall between the basal cell and the endosperm cells (Figure 7A). The cytoplasm of the chalazal suspensor cells contains mitochondria, plastids, single profiles of RER and vacuoles. Uniform distribution of single cisternae of RER is noted throughout the chalazal suspensor cells. In contrast to the basal cell, the chalazal suspensor cells contain more and bigger lipid droplets (Figure 7C,D).

In the micropylar haustorium, wall ingrowths are not observed. The cytoplasm haustorium contains a profile of RER, mitochondria, dictyosomes, vacuoles and lipid droplets (Figure 7E).

#### 2.3.6. *Echeveria lutea* (Figure 8)

The micropylar part of the basal cell wall forms wall ingrowths (Figure 8A). The cytoplasm of the basal cell is filled with mitochondria, many active dictyosomes which produce dictyosomal vesicles, plastids with electron-translucent inclusions and a bundle of microtubules (Figure 8A–D). Lipid droplets are also visible (Figure B,C). Free ribosomes are numerous and either occur singly (Figure 8C) or are aggregated in small polysomes that may have a helical arrangement (Figure 8D). The profiles of the RER are rather long and may be scattered singly throughout the cytoplasm. The RER frequently runs parallel to the plasmalemma (Figure 8B). Furthermore, the basal cell cytoplasm contains a few microbodies. They have a uniform matrix and are bounded by a single membrane. Usually, microbodies are closely associated with the cisternae RER (Figure 8B,C).

The wall separating the basal cell from the first layer of the chalazal suspensor cells contains unbranched plasmodesmata with electron-dense material (Figure 8E,F). Similar plasmodesmata with electron-dense material are also noted in the wall between the basal cell and the endosperm cells (not shown). There are notable differences between the organelles found in the basal cell and the chalazal suspensor cells. The chalazal suspensor cell plastids contain lamellae and often large starch grains. In contrast to the basal cell, the chalazal suspensor cells contain vacuoles of various sizes (Figure 8E,F).

The micropylar haustorium, forms wall ingrowths similar to those in the basal cell. The cytoplasm haustorium contains mitochondria, plastids, dictyosomes and profile RER (Figure 8G).

## 3. Discussion

### 3.1. Suspensor Diversity

A suspensor occurs in almost all angiosperm families and it is an essential organ in the development of the embryo. In different species, suspensors may have diverse size and morphology, such as filamentous (e.g., Brassicaceae), columnar, spherical or they may even be irregular in shape (as in representatives of Fabaceae) [7]. Some suspensors form single-celled or multi-nucleate haustoria which are more or less aggressive and which penetrate other integuments, endosperm and funiculus [31].

Our embryological study shows the occurrence of diverse suspensor morphologies across the two tested *Sedum* species (*S*. *sediforme* Pau and *S*. *atratum* L.) (Figure 9A,B). *S. sediforme* has a uniseriate filamentous suspensor consisting of a small haustorial basal cell and a single row of 7–10 chalazal suspensor cells (Figure 9A). Similar results have been described for *Sedum reflexum* L. (synon. *S*. *rupestre* L.) which, like *S*. *sediforme*, belongs to the same *S*. ser. *Rupestria* [20]. So far, only two species (out of seven) from the series *S*. *reflexum* [16] and *S*. *sediforme*, in this paper, (out of seven) have been studied in terms of the morphology and structure of the suspensor. In these two cases, the suspensor is a uniseriate filament of 7 to 10 cells. However, in *S. atratum* and selected species of all the other genera (*Aeonium, Monanthes, Aichryson* and *Echeveria,*) the suspensor is a few-celled multiseriate (Figure 9B–F). This organ is built of a large haustorial basal cell and a few chalazal cells in 1–2 layers similar to *S. acre* L. and *S. hispanicum* L. [22,32], and several other species of this family such as *Jovibarba sobolifera* Opiz, *Graptopetalum bellum* (Moran and J. Meyrán) D.R. Hunt and *Sempervivum arachnoideum* L. [15].

Species selected for our embryogenesis studies are classified to five other genera—*Sedum*, *Aeonium*, *Aichryson*, *Echeveria*, *Monanthes* within one subfamily Sempervivoideae [33]. Representatives of genera *Aeonium*, *Aichryson* and *Monanthes* are classified to the *Aeonium* clade, members of *Echeveria* are placed in the *Acre* clade, but species of polyphyletic genus *Sedum* are described as being representatives of the two mentioned clades and two additional ones—*Leucosedum* and *Sempervivum* [20]. Species examined during the present studies—*Aeonium sedifolium* (Webb ex Bolle) Pit. and Proust, *Aichryson laxum* (Haw.) Bramwell and *Monanthes anagensis* Praeger—were placed in the *Aeonium* clade [34]. *Sedum sediforme* is described as one of the members of the genus *Sedum* ser. *Rupestria*; however, some authors have recently separated this series to the genus level, *Petrosedum* Grulich [33]. Embryological studies point to the correctness of such a proposal [14], as do molecular phylogenetic data [20]. Formation of a uniseriate filamentous suspensor is described only in species of ser. *Rupestria* within Crassulaceae. Such morphologically different suspensors are noted in *Sedum anopetalum* DC., *S*. *reflexum* (syn. *S*. *rupestre*) and *S*. *sediforme* [14,16,32], which indicates the importance of a structural analysis of this embryonic organ. So far studies of Crassulaceae species have been carried out in selected representatives of the genera *Sedum*, *Sempervivum*, *Jovibarba* and *Graptopetalum* during embryogenesis, and these have shown both morphological and ultrastructural diversity of the suspensor cells [15,16,22,32]. Expanding knowledge about Crassulaceae embryology allows comparison of collected data with those for other (not studied) genera/species which may provide new and taxonomically useful knowledge.

### 3.2. Ultrastructure of the Suspensor

Our ultrastructural studies revealed that the micropylar end of the suspensor basal cell and the micropylar haustorium in five studied genera of Crassulaceae are covered with wall ingrowths that are typical of transfer cells. In all these cases, the wall ingrowths are surrounded by a plasma membrane, and they greatly increase the ability of these cells to absorb nutrients from the surrounding tissues. Active transportation of nutrients through the plasmalemma requires energy that is presumably provided by the many mitochondria present in transfer cells [35,36]. According to the results of our observations, the basal cell in all the studied species of the five Crassulaceae genera also have a considerable number of these organelles, which are spatially and functionally involved in the wall ingrowths at the micropylar tip of the cell. The occurrence of wall ingrowths in the suspensor cells has been described in numerous reports on many angiosperm suspensors [4,7]. Abundant dictyosomes were also detected near the developing ingrowths, for example, in *Capsella*, *Phaseolus* and *Stellaria* [7]. According to the results of our ultrastructural observations, dictyosomes were not numerous in the suspensor and haustorium of Crassulaceae near wall ingrowths. However, they were located evenly in the cytoplasm and were very active. Dictyosome vesicles may partake in the export of the material comprising wall ingrowths [37]. Moreover, the confirmed presence of many plasmodesmata crossing the transverse walls of the suspensor cells indicates the passage of nutrients from the surrounding tissues towards the embryo through the suspensor via the transfer cell [12].

The presence of various organelles in the suspensor of many flowering plants proposes that this organ may be the site of metabolism not occurring within the embryo proper [5,6,7]. A unique structural feature of suspensor cells is the presence of large, highly differentiated plastids. Specialized suspensor plastids were observed in members of many families, for example: Caryophyllaceae: *Stellaria* [38], Tropaeolaceae: *Tropaeolum* [39], Fabaceae: *Medicago* [28], *Vicia* [40], Crassulaceae: *Jovibarba*, *Sedum*, *Graptopetalum* and *Sempervivum* [41]. Most of the suspensor plastids in angiosperms typically show a dense stroma. Plastids contain internal structures that are found in no other cells of the ovule, for instance: lamellae, peripheral reticulum, plastoglobuli, tubules and electron-translucent inclusions of various sizes [31]. A unique features of suspensor plastids of three studied species (*Sedum atratum*, *Monanthes anagensis* and *Echeveria lutea* Rose) is the presence of electron-translucent inclusions. Similar structures were noted in suspensor plastids of *Stellaria media* L. [38], *Medicago sativa* L. and *M. scutellata* L. [28], *Sempervivum arachnoideum* and in endosperm chalazal haustorium *Rhinanthus serotinus* (Schönh. ex Halácsy and Heinr. Braun) Oborný (data not published). The chemical composition of the inclusions is not known. Sangduen et al. (1983) suggest that the formation and subsequent disappearance of inclusions in plastids plays a significant role in the development of the embryo and is associated with the transport of specific substances from plastids to the cytoplasm. In the studies presented here, these organelles are not so numerous in this organ but they are highly variable in shape (from spherical to more or less oval or irregular) and size. In the Crassulaceae suspensor, the plastids are in close contact with the mitochondria, endoplasmic reticulum profiles and the nucleus. A similar distribution of organelles has also been found in the suspensor cells of many species, for example: *Stellaria*, *Phaseolus Medicago* [7] and *Sedum* [16] and this suggests that this physical interaction may enhance the functional interactions between the organelles. The plastid-nucleus complex can be considered a semicell; that is, a structure capable of metabolism, photosynthesis, protein and RNA synthesis, but one which is not covered by a plasmalemma [42]. Plastids play an important role in metabolic pathways, such as starch metabolism, fatty acid and amino acid biosynthesis, aromatic and terpenoid complex production and production of phytohormones [43]. Our ultrastructural observations indicate that the suspensor and micropylar haustorium of Crassulaceae contain a moderate number of microbodies. A small number of microbodies in basal cells have also been observed for example in *Alisma plantago-aquatica* L. [44], *S*. *reflexum* [16], *Jovibarba sobolifera* and *Sempervivum arachnoideum* [15]. Some of them contain enzymes required for the conversion of fats to carbohydrates [45]. Their close association with the RER suggests that they may play active roles in cellular metabolism. The connection between the RER and microbodies may serve to enable certain integral proteins to be followed from their site of synthesis and insertion on the RER to positions in the microbody membrane. In all the studied haustorial suspensors, a significant number of cisterns of the RER were observed.

The ER is well developed in the basal cell, often running from the vicinity of the micropylar transfer wall to the chalazal part of the cell and it may well play an active part in absorption and intracellular transport of nutritive substances. Transport functions have also been proposed for the suspensor endoplasmic reticulum, for example in *Capsella* [46], *Stellaria* [38], *Phaseolus* [47], *Alisma* [44], *Jovibarba* and *Sempervivum* [15]. Biochemical analysis indicates that the ER may play a direct role in the synthesis and secretion of secondary wall materials [48].

### 3.3. Suspensor as a Connecting Structure

An important question in plant embryology is the role of communication between different parts of developing embryos and seeds of flowering plants. The presence of plasmodesmata might be a possible mode of information exchange. In a well-known *Arabidopsis* short-living suspensor, the symplasmic connectivity between the suspensor and the embryo exists at early stages of the development and disappears at heart-stage embryo [49,50]. The symplasmic barrier between the embryo and the endosperm is also formed and the transport becomes generally apoplasmic [2]. PD between suspensor and endosperm are found only in a few species (mainly among *Sedum* genus; e.g., [26]) assuming that symplasmic exchanges may occur but substantially stop early in seed development. On the contrary, our findings confirm the previously evidenced in *Sedum* [26,30] symplasmic connections of the suspensor cells both with the endosperm and the embryo proper cells (e.g., Figure 9A,B). Interestingly, the symplasmic communication exists till the end of embryogenesis and suspensor degradation.

In *Sedum*, two types of PD were found; simple, in the case of the filamentous suspensor in *S*. *sediforme*, and branched with a dome of an electron-dense material in the multiseriate suspensor of *S*. *atratum* [32]. However, in the remaining studied genera of Crassulaceae only the unbranched/branched PD with electron-dense material were found. Generally, the members of Crassulaceae show the presence of plasmodesmata in the walls between the suspensor basal cell and the endosperm. Furthermore, in *S*. *sediforme*, PD were observed in the outer walls of the chalazal suspensor cells (between the chalazal suspensor cells and the endosperm). Until now, this distribution of PD has not been observed in other species of plants. The presence of PD in the lateral walls of chalazal suspensor cells in *S*. *sediforme* could be associated with the filamentous structure of this organ. A relatively small basal cell cannot satisfy the demand of the developing embryo proper for nutrients; therefore, they are additionally transported from the endosperm. Due to the lack of literature data describing the presence of PD between the chalazal suspensor cells and the endosperm, it seems advisable to undertake further studies on the functionality of these structures. Our results confirm the PD in *S*. *atratum* and *Echeveria lutea* are generally similar in structure and distribution to those described and illustrated in *Sedum acre* and *S*. *hispanicum* [24]. Since PD with electron-dense material were noticed also in the *Sedum* embryo sac components (for details, see [51,52]), an important question, whether specific symplasmic connections in the ovule before/after fertilization are a conserved feature in angiosperms, remains to be addressed.

Typically, PD have a complex structure: they are branched and there is a conspicuous dome of electron-dense material which associates with the PD on the basal cell side. However, species from other genera of this family (e.g., *Aeonium*, *Aichryson* and *Monanthes*) have unbranched PD within a dome of electron-dense material associated with the PD on the basal cell side. Recently, Wróbel-Marek and co-workers (2017) have focused considerable attention on an investigation into plasmodesmatal permeability between the embryo/suspensor/endosperm using symplasmic transport fluorochromes. Experiments have confirmed the functionality of the compound plasmodesmata, but only in the directions of the basal cell-embryo proper and basal cell-endosperm. No movement of symplasmic transport fluorochromes in the opposite direction was observed. Unidirectional movement through plasmodesmata was not irrefutably proven, but data revealed such a possibility. One of the key questions concerns the chemical composition of the dome of electron-dense material. These investigations are in progress.

### 3.4. Suspensor Specialization

The results of the present study revealed that in all species of Crassulaceae, enlargement of the basal cell nucleus is one of the first indications of its specialization. Many angiosperms show a high degree of ploidy during suspensor differentiation: *Phaseolus coccineus* L. 8192C [53], *Triglochin palustre* L. 256C [54] and *Sedum acre* 1024C [21].

An interesting phenomenon occurring in the studied *S*. *sediforme* suspensor is the insertion of an enlarged nucleus from basal cell to micropylar haustorium. We consider three explanations for this phenomenon: (i) the basal cell nucleus is shifted towards the greater mass of the cytoplasm of the cell where more intense metabolic processes occur; (ii) the large polyploid basal cell nucleus is pushed out under pressure from a relatively small basal cell in the studied species; (iii) the process is associated with a specific interaction of individual components of the cytoskeleton. Our ultrastructural study indicated that in the cytoplasm of the basal cell of tested species, especially in *S. atratum*, *A*. *sedifolium*, *M*. *anagensis* and *E*. *lutea*, tubulin cytoskeleton was visible. Microtubules showed longitudinal or transverse distribution along the long axis of the cell, and some microtubular bundles appeared to pass close to the surface of the nucleus. The cytoskeleton plays an important role in many cellular processes, for example cellular signaling, organelle motility and subcellular compartmentation during plant growth and development [55,56]. To date, only a few studies have been reported on the cytoskeleton during the formation, development and differentiation of polyploid cells of embryo-suspensor in: *Sedum acre* [23], *Alisma plantago-aquatica* [57] and *Gagea lutea* (L.) Ker Gawl., [58]. In all of the above species, the authors have shown the presence of an abundant actin and tubulin cytoskeleton in endopolyploid suspensor basal cells. According to these authors, the presence of an abundant cytoskeleton is probably associated with the positioning of the huge nucleus, movement of organelles and transport of nutrients within the metabolically active basal cell to the developing embryo proper. There are several reports in the literature which have indicated that the massive suspensors of *Phaseolus coccineus* and *Tropaeolum majus* L. may be more involved in macromolecular synthesis than smaller suspensors are, and they may serve as storage tissue that provides nutritional support for embryogenesis of the developing late-stage embryo [31], while more reduced filamentous suspensors (as for example in *Arabidopsis* or the here studied *Sedum atratum*) may function primarily in absorbing nutrients from maternal tissues and transporting materials to the embryo proper [4].

## 4. Materials and Methods

### 4.1. Plant Material

Flowering plants of the genus *Sedum* (*S*. *atratum* L. voucher number D.M.–2013/001 and *S*. *sediforme* Pau voucher number D.M.–2013/002) were sourced from the Botanic Garden of the Jagiellonian University, Kraków, Poland. Flowers of other genera (*Aeonium*: *A*. *sedifolium* (Webb ex Bolle) Pit. & Proust voucher number M.K-K–2015/001, *Aichryson*: *A*. *laxum* (Haw.) Bramwell voucher number M.K-K–2015/002, *Echeveria*: *E*. *lutea* Rose voucher number M.K-K–2015/003 and *Monanthes*: *M*. *anagensis* Praeger voucher number M.K-K–2015/004) were collected from the Prague Botanical Garden, Czech Republic. All voucher specimens are deposited in the Herbarium of the Department of Plant Cytology and Embryology, Faculty of Biology, University of Gdańsk, Poland.

### 4.2. Electron Microscopy

Ovules were obtained from the flowers, and these ovules at various stages of embryo development were fixed in 2.5% (*w*/*v*) formaldehyde (freshly obtained from paraformaldehyde) and 2.5% (*v*/*v)* glutaraldehyde in a 0.05 M cacodylate buffer (pH = 7.0) for 4 h at room temperature. The ovules were then rinsed in the same buffer and post-fixed in 1% (*w*/*v*) osmium tetroxide in a cacodylate buffer at 4 °C overnight, and were processed with 1% (*w*/*v*) uranyl acetate in distilled water for 1 h, dehydrated in a graded acetone series and then embedded in Spurr’s epoxy resin [59]. Ultrathin (60–100 nm) section series of the embedded material were obtained using a diamond knife on a Leica EM UC7 ultramicrotome and then post-stained with a saturated solution of uranyl acetate in 50% (*w*/*v*) ethanol and with 0.04% (*w*/*v*) lead citrate [60]. Observations were made using a Philips CM 100 at 80 kV and an FEI Tecnai G2 Spirit TWIN/BioTWIN transmission electron microscope at 120 kV at the Laboratory of Electron Microscopy, Faculty of Biology, University of Gdańsk (Gdańsk, Poland).

### 4.3. Light Microscopy

For light microscopy, ovules were fixed and embedded as described for ultrastructural studies. Semi-thin sections (0.5–1 μm) were cut with glass knives on a Leica EM UC7 ultramicrotome and mounted on glass slides. The material was subsequently placed on a hot plate and stained with 0.05% (*w*/*v*) Toluidine Blue O (TBO) for 1 min at 60 °C. Microscopic analysis and photographic observations were made using a Nikon Eclipse E 800 light microscope and a Nikon DS-5Mc camera, respectively, with Lucia Image software.

## 5. Conclusions

The results indicated that the suspensor in selected species and/or genera of Crassulaceae is a dynamic part of the embryo complex, functioning in absorption, short-distance translocation and exchange of metabolites needed in early stages of embryogenesis. The suspensor of these all species sustains symplasmic connections (via PD) not only between its cells and the embryo proper but also between the endosperm, and additionally builds apoplasmic connection with the ovule tissue (by haustoria and transfer walls) until the suspensor degradation. Morphological and ultrastructural observations suggest that particularly in *Sedum* the developmental pattern of the suspensor influences the diversity in the plasmodesmata structure. Two types of PD were found here: simple PD in *S*. *sediforme* (as a consequence of uniseriate filamentous suspensor occurring), and branched PD with an electron-dense material in a multiseriate suspensor of *S*. *atratum*. One type of PD but the same dependence occurs in the rest of the tested genera (*Aeonium*, *Aichryson*, *Monanthes* and *Echeveria*) —the multiseriate suspensor with unbranched/branched electron-dense PD (summarized in Figure 9 and Table 1).

The plasmodesmata diversity phenomenon in Crassulaceae may be an evolutionary adaptation regulating the flow of nutritional substances to the embryo. Future studies will be focused on the identification of components of electron-dense material associated with plasmodesmata using electron immunogold labeling. New insights into the plasmodesmatal structure/function relationships of Crassulacean suspensors will be gained through advances in our understanding of complex plasmodesmata. Furthermore, this new knowledge will provide many features that can potentially be used to trace and understand the evolution of these plasmodesmata in the ovule.

## Figures and Tables

**Figure 1 plants-09-00320-f001:**
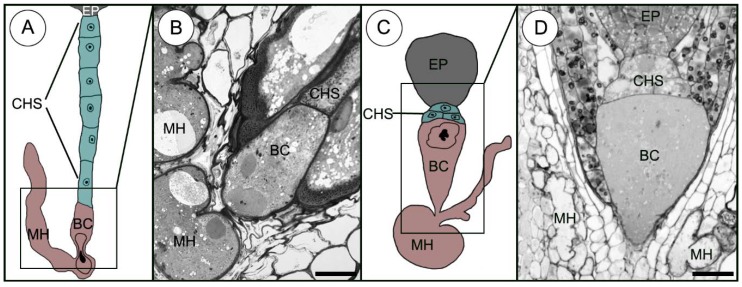
Two types of suspensor development in *Sedum*. The *Sedum* suspensor. (**A**) Drawings illustrating a uniseriate suspensor in *S*. *sediforme*. (**B**) Longitudinal semi-section. (**C**) Drawings illustrating a few-celled multiseriate suspensor in *S*. *atratum*. (**D**) Longitudinal semi-section. BC—basal cell, CHS—chalazal suspensor cells, EP—embryo proper, MH—micropylar haustorium. Scale bars: **B** = 10 μm; **D** = 25 μm.

**Figure 2 plants-09-00320-f002:**
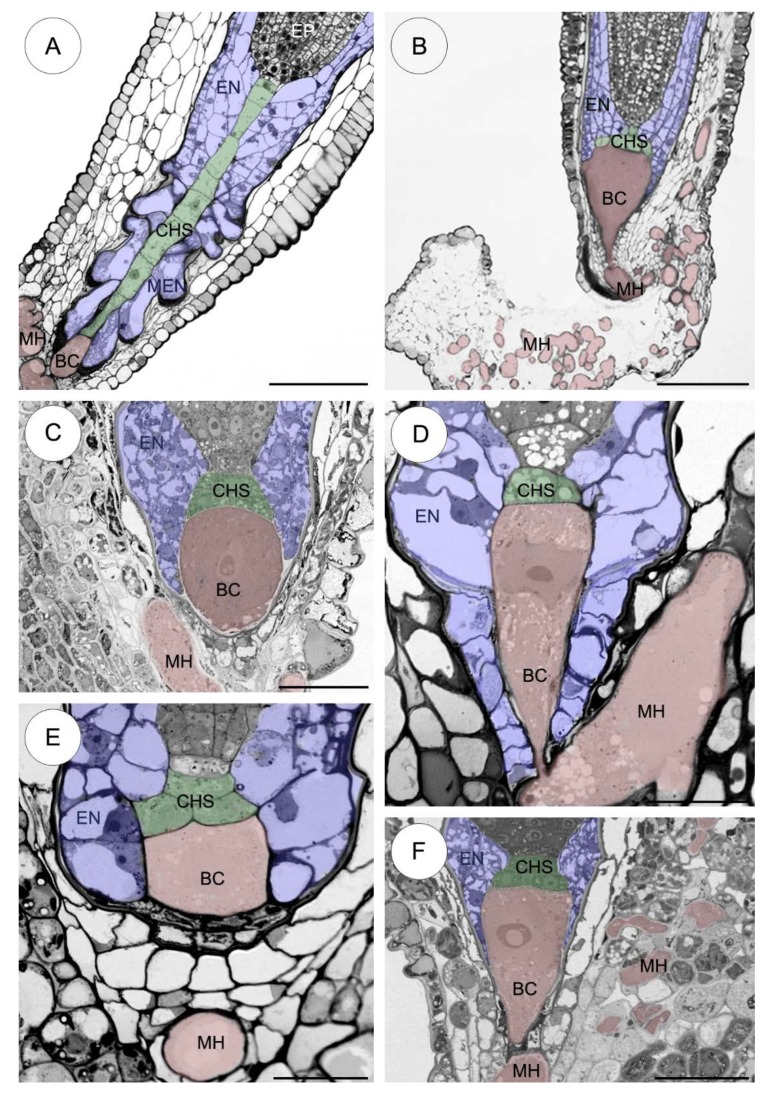
Haustorial suspensor morphology in representatives of Crassulaceae genera. (**A**–**F**) Light micrographs. (**A**) Long uniseriate suspensor consisting of a basal cell forms haustorium (red) and chalazal suspensor cells (green) in *Sedum sediforme*. (**B**–**F**) Few-celled multiseriate suspensor consisting of a basal cell forms haustorium (red) and chalazal suspensor cells in 1–2 layers in *Sedum atratum* (**B**), *Aeonium sedifolium* (**C**), *Monanthes anagensis* (**D**), *Aichryson laxum* (**E**), *Echeveria lutea* (**F**), respectively. BC—basal cell, CHS—chalazal suspensor cells, EN—endosperm (blue), EP—embryo proper, MH—micropylar haustorium, MEN—micropylar endosperm (blue). Scale bars: **A**, **B** = 100 μm; **C**, **F** = 50 μm; **D**, **E** = 20 μm.

**Figure 3 plants-09-00320-f003:**
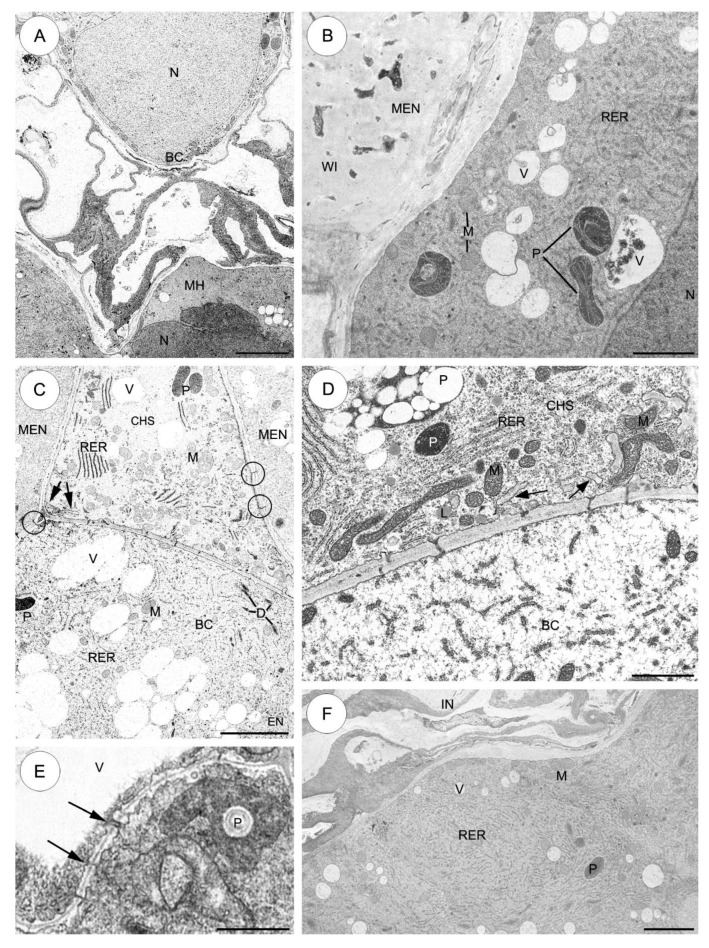
The suspensor cells of *Sedum sediforme*. **A**–**F**, Electron micrographs. (**A**) The micropylar part of the basal cell with a fragment of the micropylar haustorium. The nucleus is visible in the basal cell. A higher magnification of the basal cell and the micropylar haustorium. Note pushing of the nucleus from the basal cell towards the micropylar haustorium. (**B**) Higher magnification of a portion of a basal cell in the vicinity of the nucleus. Cytoplasm is rich in mitochondria, short profiles of rough endoplasmic reticulum, enlarged electron-dense plastids, mitochondria and different size vacuoles. The micropylar endosperm walls form an extensive network of ingrowths. (**C**) Part of cell wall between basal cell and first cell of chalazal suspensor cells with simple plasmodesmata and fine wall ingrowths (arrows). In chalazal region of the basal cell, note a group of vacuoles containing fibrillar material appears together with mitochondria, a few dictyosomes, short profiles of rough endoplasmic reticulum, occasionally plastids. Lateral walls of the suspensor cells on the border with the endosperm have simple plasmodesmata (rings). (**D**) Higher magnification fragment of cell wall between basal cell and first cell of chalazal suspensor cells with plasmodesmata and fine wall ingrowths (arrows); lipid droplet, mitochondrion, plastid, rough endoplasmic reticulum. (**E**) Portion of chalazal suspensor cells. Plasmodesmata (arrows) present in cell wall separating chalazal suspensor cells; plastid, vacuole. (**F**) Fragment of the micropylar haustorium. Note the numerous profiles of rough endoplasmic reticulum, small vacuoles and mitochondria. BC—basal cell, CHS—chalazal suspensor cells, D—dictyosome, EN—endosperm, INT—integument, L—lipid droplet, M—mitochondrion, MEN—micropylar endosperm, MH—micropylar haustorium, N—nucleus, P—plastid, RER—rough endoplasmic reticulum, V—vacuole, WI—wall ingrowths. Scale bars: **A**, **C**, **F** = 5 μm; **B**, **D**, **E** = 2 μm.

**Figure 4 plants-09-00320-f004:**
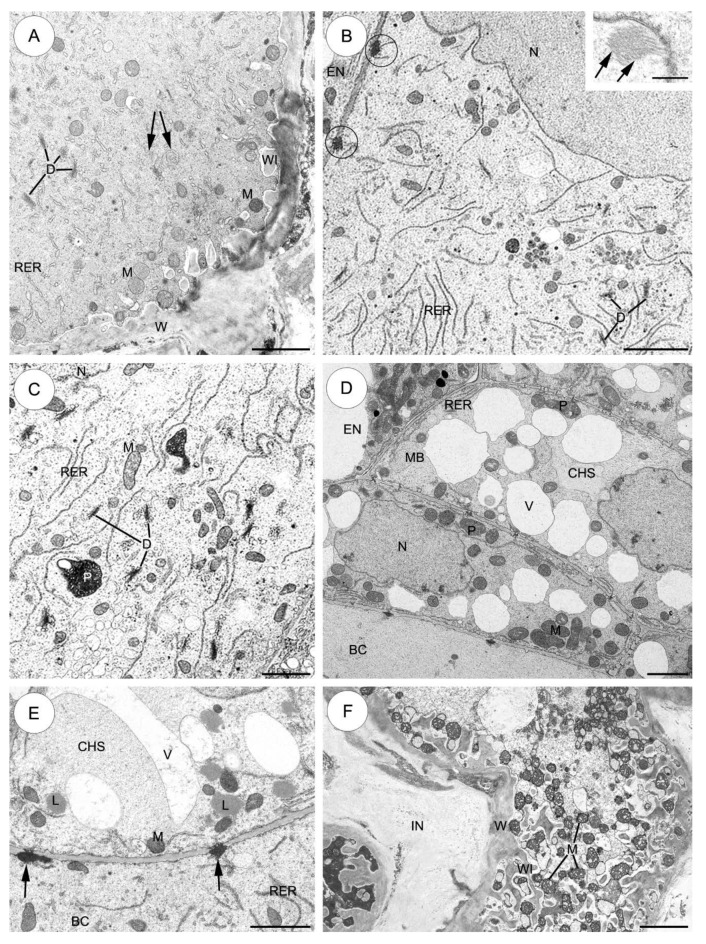
The suspensor cells of *Sedum atratum*. (**A**–**F**) Electron micrographs. (**A**) The micropylar part of the basal cell shows wall ingrowths formed along the basal wall. The cytoplasm of the basal cell has abundant mitochondria, profiles of endoplasmic reticulum and dictyosomes; myelin-like bodies (arrows). (**B**) Fragment of the large basal cell in the vicinity of the nucleus bordered by endosperm. The electron-dense material associated with the branched plasmodesmata (rings) can be observed. Cytoplasm is filled with very numerous long profiles of rough endoplasmic reticulum, dictyosomes and mitochondria; inset: Note the bundles of microtubules (arrows) near the nucleus. (**C**) Higher magnification of a portion of the basal cell in the vicinity of the nucleus. Cytoplasm is rich in mitochondria, profiles of rough endoplasmic reticulum, active dictyosomes and a few electron-dense plastids. (**D**) Plasmodesmata are presented between the individual cells and in walls separating the suspensor from the rest of the embryo. The chalazal cells of the suspensor are more strongly vacuolized than the basal cell; nucleus. (**E**) High magnification of the wall separating the basal cell from the first layer of the chalazal suspensor cells. Note the numerous branched plasmodesmata with electron-dense material associated on the cell side (arrows); lipid droplets, profiles of rough endoplasmic reticulum, vacuoles. (**F**) Portion of the micropylar haustorium branching into integumentary tissues. Note the well-developed wall ingrowths and the numerous mitochondria near the wall ingrowths. BC—basal cell, CHS—chalazal suspensor cells, D—dictyosome, EN—endosperm, ER—endoplasmic reticulum, IN—integument, L—lipid droplet, M—mitochondrion, MB—microbody, MH—micropylar haustorium, N—nucleus, P—plastid, RER—rough endoplasmic reticulum, V—vacuole, WI—wall ingrowths, W—cell wall. Scale bars: **A**, **D** = 2 μm; **B** = 5 μm; inset 500 nm; **C**–**F**= 2.5 μm; **E**= 1.5 μm.

**Figure 5 plants-09-00320-f005:**
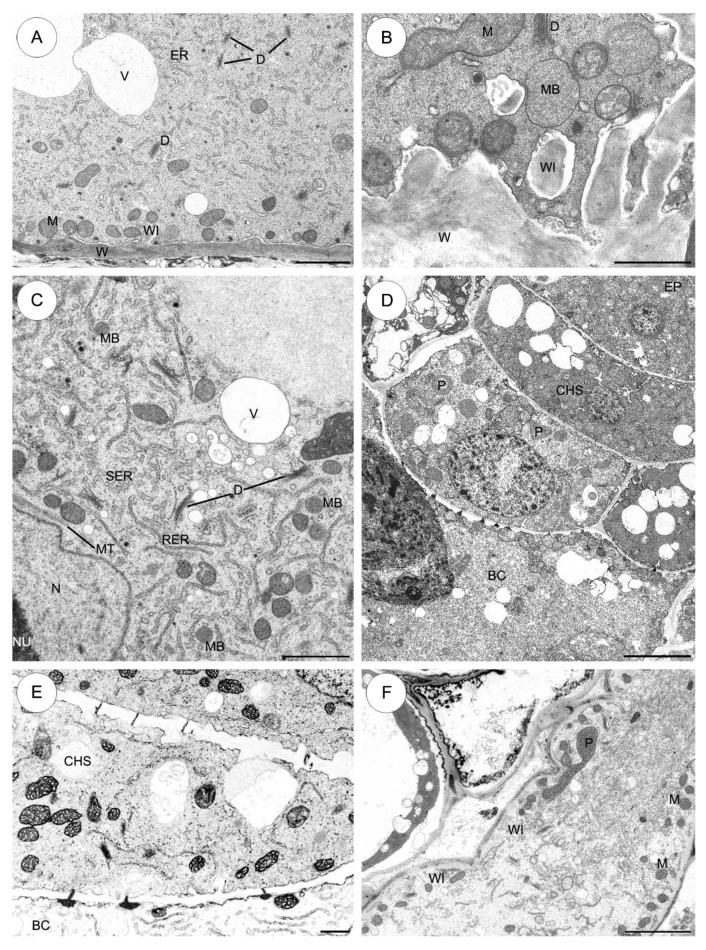
The suspensor cells of *Aeonium sedifolium*. (**A**–**F**) Electron micrographs. (**A**) A part of the micropylar apex of the basal cell shows wall ingrowths formed along the basal wall. The cytoplasm of the basal cell has abundant mitochondria, short profiles of endoplasmic reticulum, dictyosomes and vacuoles. (**B**) Higher magnification of fragment of the micropylar apex of the basal cell showing wall ingrowths and basal cell wall. Note the mitochondria near the wall ingrowths, microbodies and dictyosomes. (**C**) Part of the basal cell in the vicinity of the nucleus. Cytoplasm is filled with rough endoplasmic reticulum cisternae, a well-developed smooth endoplasmic reticulum and dictyosomes, plastids, small vacuoles and microbodies; nucleolus. Note the bundles of microtubules near the nucleus. (**D**) Chalazal part of the basal cell with chalazal suspensor cells and a few cells of the embryo proper are visible. (**E**) Plasmodesmata in the suspensor cells. Unbranched plasmodesmata with electron-dense material located on the cytoplasm side perforate the wall separating the basal cell from the first layer of the chalazal suspensor cells. Longitudinal wall between the cells inside the first layer of the chalazal suspensor. Note the numerous typical plasmodesmata. (**F**) Portion of the micropylar haustorium. Note the mitochondria between wall ingrowths and a few electron-dense plastids. BC—basal cell, CHS—chalazal suspensor cells, D—dictyosome, EP—embryo proper, ER—endoplasmic reticulum, M—mitochondrion, MB—microbody, MH—micropylar haustorium, MT—microtubules, N—nucleus, NU—nucleolus, P—plastid, RER—rough endoplasmic reticulum, SER—smooth endoplasmic reticulum, V—vacuole, WI—wall ingrowths, W—cell wall. Scale bars: **A** = 2 μm; **B**, **D**, **E** = 1 μm; **C** = 2 μm; **E** = 1 μm; **F** = 5 μm.

**Figure 6 plants-09-00320-f006:**
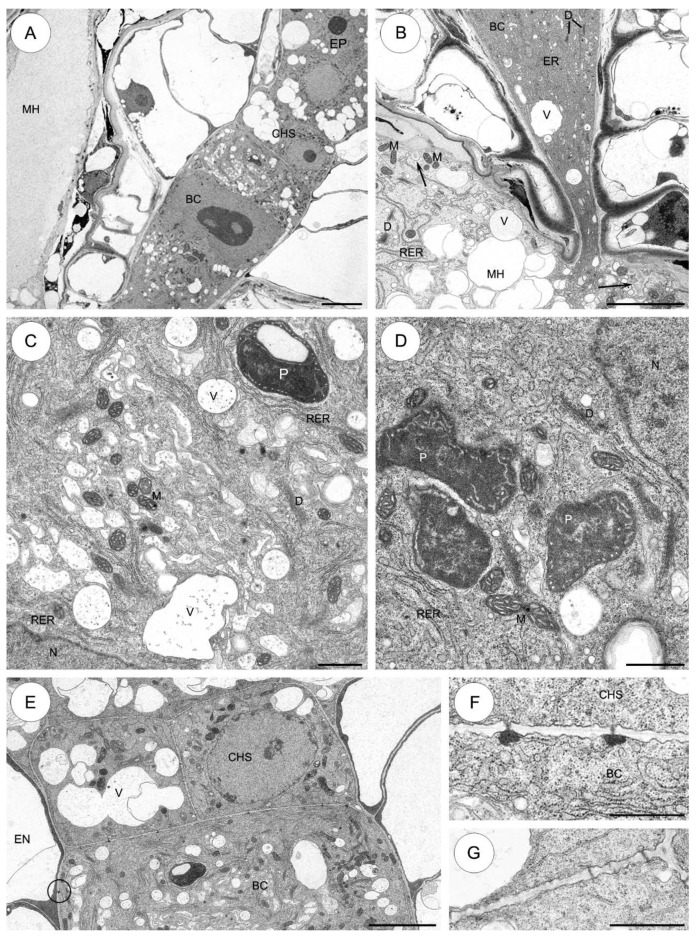
The suspensor cells of *Monanthes anagensis*. (**A**–**G**) Electron micrographs. (**A**) Electron micrograph showing the elongated basal cell with the strongly developed micropylar haustorium and a few chalazal cells in one layer. (**B**) The tip of the basal cell with a well-developed micropylar haustorium. Prior to intruding through the integument, the tip of the basal cell shows a very dense cytoplasm with few organelles, and the basal cell haustorium is filled with electron-loose cytoplasm containing ribosomes, mitochondria, giant plastids, profiles of the rough endoplasmic reticulum, microbodies and vacuoles. The micropylar haustorium wall forms delicate ingrowths (arrows). (**C**) Higher magnification of part of basal cell in the vicinity of the nucleus. Cytoplasm is rich in mitochondria, profiles of rough endoplasmic reticulum, active dictyosomes and many different-sized vacuoles with fibrillar material. Occasionally, large electron-dense plastids with electron-transparent inclusions were observed. (**D**) Fragment of cytoplasm of the basal cell. Note the enlarged plastids, mitochondria, profile of the RER, dictyosomes and vacuoles. (**E**) The chalazal suspensor cells linking the basal cell with the embryo proper. Note the vacuoles which are larger than in the basal cell. Note plasmodesmata with electron-dense material in the wall between the basal cell and the endosperm cells (ring). (**F**) High magnification of the wall separating the basal cell from the first layer of the chalazal suspensor cells. Plasmodesmata with electron-dense material are noted. (**G**) A longitudinal wall between the cells inside the first layer of the chalazal suspensor. Numerous typical plasmodesmata are seen. BC—basal cell; CHS—chalazal suspensor cells; D—dictyosome, EN—endosperm; ER—endoplasmic reticulum, M—mitochondrion, MH—micropylar haustorium, N—nucleus, P—plastid, RER—rough endoplasmic reticulum, V—vacuole. Scale bars: **A**, **B**, **E** = 5 μm; **C**, **D**, **F**, **G** = 1 μm.

**Figure 7 plants-09-00320-f007:**
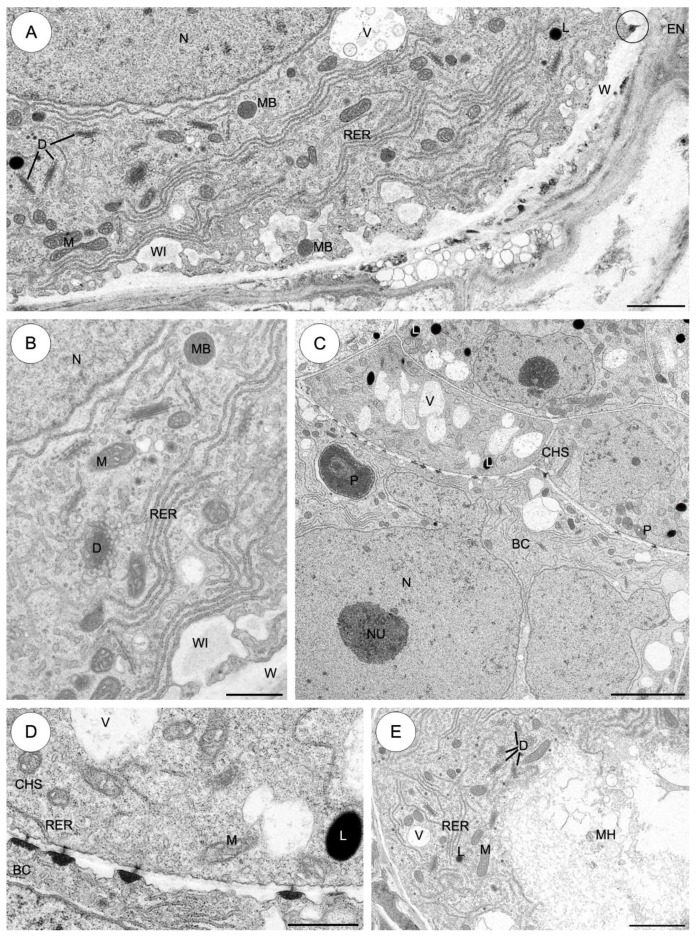
The suspensor cells of *Aichryson laxum*. (**A**–**E**) Electron micrographs. (**A**) The micropylar part of the basal cell shows wall ingrowths formed along the basal wall. Plasmodesmata are observed in the wall between the basal cell and the endosperm (ring). The cytoplasm of the basal cell has abundant mitochondria, profiles of endoplasmic reticulum, vacuoles, dictyosomes, microbodies and some lipid droplets. Nucleus is situated in the central part of the basal cell. (**B**) Higher magnification of part of the basal cell in the vicinity of the nucleus. Note the numerous profiles of endoplasmic reticulum, mitochondria, active dictyosomes, microbodies and small vacuoles. A portion of the wall ingrowths is visible. (**C**) Fragment of the chalazal part of the basal cell with the nucleus and nucleolus, and chalazal suspensor cells are visible. In the chalazal cells, lipid droplets and groups of vacuoles are visible; plastid. (**D**) Plasmodesmata in the suspensor cells. Plasmodesmata with electron-dense material are observed in the wall separating the basal cell from the first layer of the chalazal suspensor cells; profiles of rough endoplasmic reticulum, vacuoles, mitochondria and lipid droplets. (**E**) Fragment of the micropylar haustorium. Note the numerous mitochondria, profiles of rough endoplasmic reticulum, dictyosomes, vacuoles and some lipid droplets. BC—basal cell, CHS—chalazal suspensor cells, D—dictyosome, EP—embryo proper, L—lipid droplet, M—mitochondrion, MB—microbody, MH—micropylar haustorium, N—nucleus, NU—nucleolus, P—plastid, RER—rough endoplasmic reticulum, V—vacuole, WI—wall ingrowths, W—cell wall. Scale bars: **A**= 2.5 μm, **B**–**D** = 1 μm; **E** = 2.5 μm.

**Figure 8 plants-09-00320-f008:**
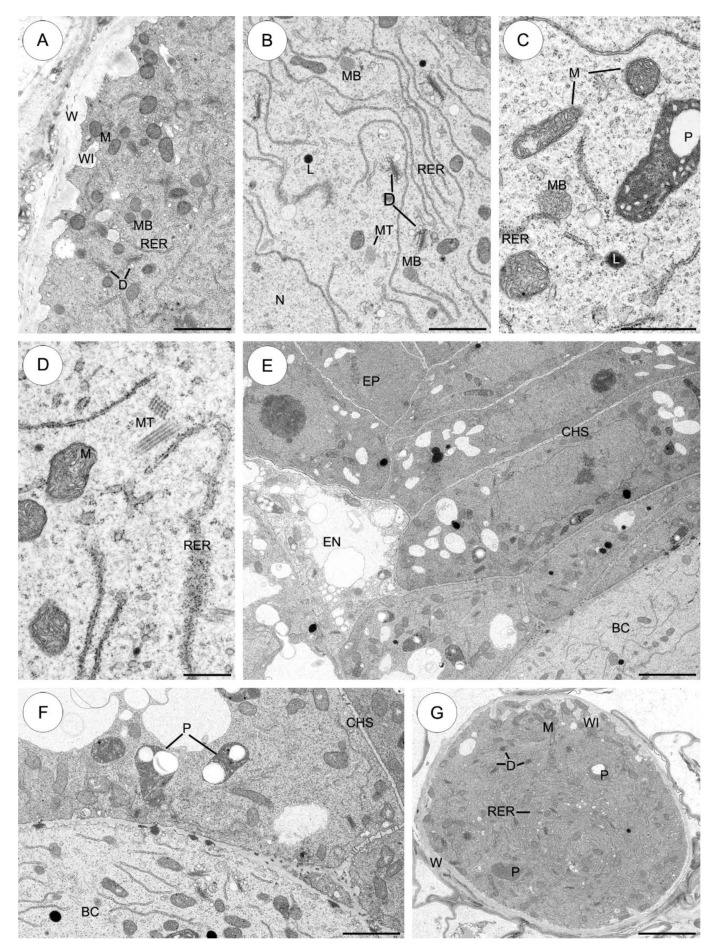
The suspensor cells of *Echeveria lutea*. (**A**–**G**) Electron micrographs. (**A**) The micropylar part of the basal cell shows wall ingrowths formed along the basal wall. The cytoplasm of the basal cell has abundant mitochondria, single profiles of endoplasmic reticulum, dictyosomes and microbodies. (**B**) Higher magnification of part of the basal cell in the vicinity of the nucleus. Cytoplasm is rich in mitochondria, long profiles of rough endoplasmic reticulum, active dictyosomes, microbodies and a bundle of microtubules. (**C**,**D**) Fragments of cytoplasm of the basal cell with visible organelles: mitochondria, profiles of rough endoplasmic reticulum and bundles of microtubules. The plastid with an electron-translucent inclusion is seen. (**E**) Chalazal suspensor cells connecting the basal cell with the embryo proper. (**F**) Plasmodesmata in the suspensor cells. Plasmodesmata present in the cell wall separating chalazal suspensor cells. The cytoplasm of the chalazal suspensor cells has of a few starch-containing plastids. (**G**) Fragment of the micropylar haustorium. Note the mitochondria between wall ingrowths and dictyosomes, profiles of rough endoplasmic reticulum and a few prominent plastids occasionally with electron-translucent inclusions. BC—basal cell, CHS—chalazal suspensor cells, D—dictyosome, EN—endosperm, EP—embryo proper, L—lipid droplet, M—mitochondrion, MB—microbody, MH—micropylar haustorium, MT—microtubules, P—plastid, RER—rough endoplasmic reticulum, V—vacuole, WI—wall ingrowths, W cell wall. Scale bars: **A**, **B**, **F**, **H** = 2 μm; **C** = 1 μm; **D** = 0.5 μm. **E**, **G** = 5 μm.

**Figure 9 plants-09-00320-f009:**
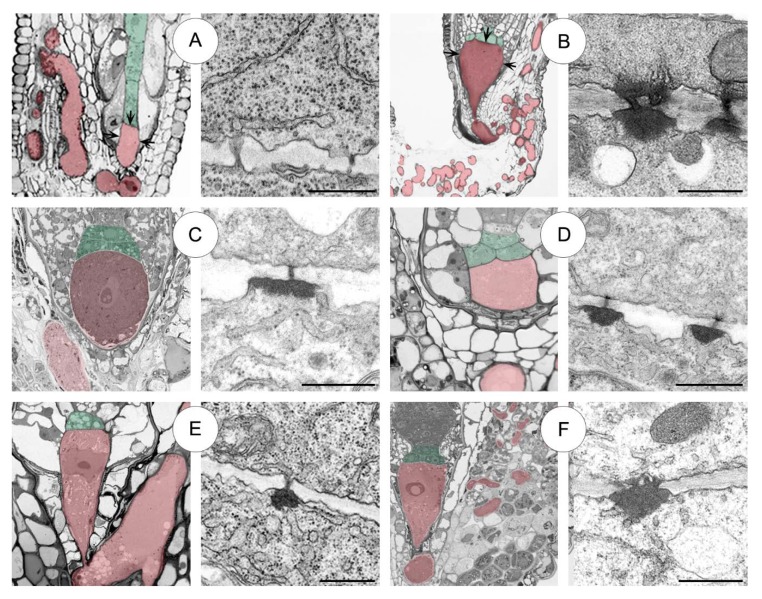
Suspensor types, plasmodesmata distribution and ultrastructure in representatives of five Crassulaceae genera. (**A**) *Sedum sediforme*. (**B**) *S*. *atratum*. (**C**) *Aeonium sedifolium*. (**D**) *Aichryson laxum*. (**E**) *Monanthes anagensis*. (**F**) *Echeveria lutea*. Haustorial basal cells (red) and chalazal suspensor cells (green) are highlighted (see left column), and plasmodesmata (see right column). Arrows in (**A**) and (**B**) show the distribution of plasmodesmata in the suspensor. Scale bars: **A** = 250 nm; **B**, **C**, **E**, **F** = 500 nm; **D** = 1 μm.

**Table 1 plants-09-00320-t001:** Suspensor and plasmodesmata types in selected representatives of the genera *Sedum*, *Aeonium*, *Aichryson*, *Monanthes* and *Echeveria*.

Suspensor’s Characters		*Sedum*		*Aeonium*	*Aichryson*	*Monanthes*	*Echeveria*
		*S. sediforme*	*S. atratum*	*A. sedifolium*	*A. laxum*	*M. anagensis*	*E. lutea*
Type of Suspensor	A long uniseriate	+	-	-	-	-	-
	A few-celled multiseriate	-	+	+	+	+	+
Type of Plasmodesmata	Simple	+	-	-	-	-	-
	Unbranched/branched with an electron-dense material	-	+	+	+	+	+
Symplasmic connection via PD between suspensor and embryo and between suspensor and endosperm		Present in all genera/species					
Apoplasmic connection with the ovule tissue (by haustoria and transfer cells)		Present in all genera/species					

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
