# Peer review of "Development of Embryo Suspensors for Five Genera of Crassulaceae with Special Emphasis on Plasmodesmata Distribution and Ultrastructure"

_plants, 2020, doi:10.3390/plants9030320_

Round 1

Reviewer 1 Report

The work extends microscopic, qualitative observations of suspensor morphology to a much more detailed, ultrastructural viewpoint. The value of the work is connected to possible evolutionary interpretations of embryo development and to extend the number of plant families that have such detailed descriptions.

The discussion and description of light and electron micrographs is robustly detailed. The text of this manuscript is very dense and an interested but unsophisticated reader may have difficulty connecting the discussion of Figs 3-8. A summary Table might help, but as this reviewer is among the unsophisticated the discussion section may be sufficient and I would hesitate to add content here.

The work is comprehensive, detailed and advances the understanding of embryo development in general and suspensor attributes unique to the genera studied.

Reviewer 2 Report

The manuscript is excellent. It is well prepared and written. It is an extensive study. The findings are interesting and new. Language is perfect. I only have three minor suggestions:

Keywords: Please do not repeat keywords from the title.

Line 461: Please provide examples of botanical families/genera for every suspensor shape (now there are only two)

In the Discussion, I suggest providing full botanical names of species when mentioned the first time, including the initial of the Autor(s) who first validly published a botanical name, e.g. Tropaeolum majus L.
